# MicroRNA-Enriched Exosomes from Different Sources of Mesenchymal Stem Cells Can Differentially Modulate Functions of Immune Cells and Neurogenesis

**DOI:** 10.3390/biomedicines10010069

**Published:** 2021-12-30

**Authors:** Naina Soni, Suchi Gupta, Surender Rawat, Vishnu Krishnakumar, Sujata Mohanty, Arup Banerjee

**Affiliations:** 1Laboratory of Virology, Regional Centre for Biotechnology, Faridabad 121001, India; naina.soni@rcb.res.in (N.S.); surender.rawat@rcb.res.in (S.R.); 2DBT-Centre of Excellence for Stem Cell Research, Stem Cell Facility, All India Institute of Medical Sciences, New Delhi 110029, India; gupta.s1291@gmail.com (S.G.); kvishnu299@gmail.com (V.K.)

**Keywords:** mesenchymal stem cells, exosomes, miRNA, neutrophils, PBMC, neurogenesis

## Abstract

Adult Mesenchymal stem cells-derived exosomes carry several biologically active molecules that play prominent roles in controlling disease manifestations. The content of these exosomes, their functions, and effect on the immune cells may differ depending on their tissue sources. Therefore, in this study, we purified the exosomes from three different sources and, using the RNA-Seq approach, highly abundant microRNAs were identified and compared between exosomes and parental cells. The effects of exosomes on different immune cells were studied in vitro by incubating exosomes with PBMC and neutrophils and assessing their functions. The expression levels of several miRNAs varied within the different MSCs and exosomes. Additionally, the expression profile of most of the miRNAs was not similar to that of their respective sources. Exosomes isolated from different sources had different abilities to induce the process of neurogenesis and angiogenesis. Moreover, these exosomes demonstrated their varying effect on PBMC proliferation, neutrophil survival, and NET formation, highlighting their versatility and broad interaction with immune cells. The knowledge gained from this study will improve our understanding of the miRNA landscape of exosomes from hMSCs and provide a resource for further improving our understanding of exosome cargo and their interaction with immune cells.

## 1. Introduction

Human mesenchymal stem cells (hMSCs) are widely recognized for their regenerative nature and studied as therapeutic modalities in many chronic and inflammatory diseases. They are multilineage progenitor cells, well known for their distinct role in repair, regenerative mechanisms [1,2,3], and regulating immune responses [4,5,6]. Several studies have suggested that hMSCs exist in many tissues such as adipose tissue, bone marrow, salivary glands, Wharton jelly, and dental pulp [7] and can be used as therapeutic interventions. By September 2021, more than 1300 MSC-related clinical trials were registered on the NIH Clinical Trial Database (https://clinicaltrials.gov/, accessed on 22 September 2021). Many of the published shreds of evidence have focused on the secretory proteins such as cytokines, chemokine, growth factors, and extracellular vesicles (EVs) as the critical drivers of their functional effects. Most of the properties of hMSCs can be recapitulated through a particular class of extracellular vesicles called exosomes [8]. Exosomes are vesicles of endocytic origin with a size range between 30–150 nm [9]. The content of exosomes mostly mimics that of the cells from which they originated. Exosomal membranes are rich in signaling molecules, surface antigens, proteins, and genetic material [10,11]. Exosomes derived from hMSCs have been shown to enhance neurovascular plasticity after stroke [12], dormancy in metastatic breast cancer cells [13], skeletal muscle regeneration [14], cutaneous wound healing [15], and cartilage regeneration [16]. This range of exosomal therapeutic potential and efficacy supports its candidacy for one of the best therapeutics to treat a broad spectrum of diseases compared to the use of relatively large living hMSCs. Transplantation of living replicative cells is also risky as the grafted cells cannot be removed in adverse response events or upon disease resolution.

In contrast with large living cells, nano-sized and non-living exosomes would not occlude microvasculature, transform into inappropriate cell types, or persist as permanent grafts upon cessation of therapies. Overall, hMSCs-derived, exosome-based therapeutics could be a safer, cheaper, and more effective treatment modality than cell-based MSC therapeutics. About 72% of the studies suggested that hMSCs-derived exosomes are considered safe for cell-free therapeutics [17]. Exosomes purified from various hMSCs are widely studied. Since the content of exosomes may differ based on their sources, it is essential to understand the bioactive molecules present in the exosomes. Apart from proteins, exosomes are also loaded with miRNAs that have gained much attention recently. The miRNAs from exosomes have been exploited to understand the mechanisms of their regenerative and immunomodulatory roles [18,19,20,21,22].

Here, we performed an in-depth characterization of hMSC exosomes derived from three different sources, i.e., bone marrow (BM), adipose tissue (AD), and Wharton Jelly (WJ), and compared them with cellular miRNAs. This study aimed to get more in-depth insights into the differences and similarities between exosomes isolated from different origins in the context of their miRNA landscape and to study the differences in immunomodulatory roles of these exosomes from different sources. Our analysis of the miRNA landscape of exosomes and their immunomodulatory roles is useful in understanding the effect of these exosomes on immune cells, which may have consequences in their therapeutic usage.

## 2. Material and Methods

### 2.1. Ethics Approval and Consent to Participate

This study was initiated after obtaining ethical clearance from the Institutional Committee for Stem Cell Research (ICSCR), All India Institute of Medical Sciences, New Delhi, India, Reference No. IC-SCR/55/16(R). All the samples were obtained after taking the donors’ consent according to the guidelines of the ethical committee. All the blood-related experiments were also performed with proper donors’ consent and after approval from the Institutional Ethical Committee (Human Research) Regional Centre for Biotechnology, Faridabad, Haryana, India, Reference no: RCB-IEC-H-25.

### 2.2. Isolation and Culture of hMSCs from Bone Marrow (BM), Adipose Tissue (AD), and Wharton Jelly (WJ)

#### 2.2.1. Isolation and Culture of Bone Marrow-Derived Stem Cells (BM-hMSCs)

Bone marrows were collected from the donors (*n* = 4) undergoing the routine medical test procedure at AIIMS, New Delhi. The hMSCs were isolated and cultured, as previously described [23]. Briefly, neat bone marrows were seeded in a 60-mm culture dish (BD, Franklin Lakes, NJ, USA) in complete growth media containing 1X Dulbecco’s Modified Eagle Medium–Low Glucose (DMEM-LG) (Life Technologies, Carlsbad, CA, USA) media with 10% Fetal Bovine Serum (FBS) (HyClone, Logan, UT, USA) and 1% Penicillin/Streptomycin (PS) (Life Technologies, Carlsbad, CA, USA). The cells were incubated in a CO_2_ incubator. The media were changed every third day until the cell confluency reached 80%. Adherent cells were then passaged with 0.05% trypsin-EDTA (Invitrogen-Gibco, Waltham, MA, USA) and reseeded at a seeding density of 1 × 10^4^ cells/cm^2^ for further experiments.

#### 2.2.2. Adipose Tissue-Derived Mesenchymal Stem Cells (AD-hMSCs)

Adipose tissues were collected from the donors (*n* = 4) undergoing a pre-scheduled surgical procedure at AIIMS, New Delhi. The samples were collected in a 5-mL transport vial containing DMEM-LG without FBS with 1% PS and Gentamycin (250 μg/mL). The samples were washed with PBS containing 1% PS and Gentamycin (250 μg/mL). The explants (~2 mm) were then carefully placed in a 35-mm culture dish and kept undisturbed in a CO_2_ incubator overnight. As the tissues started adhering to the surface on the next day, complete media were added and changed every 3 to 4 days. After the cells started growing and migrating out of the explant, and reached 80% confluence, they were harvested using 0.05% trypsin-EDTA (Invitrogen-Gibco, Waltham, MA, USA) and transferred into a 60-mm culture dish for further experiments. All assays were performed using hMSCs at passage 3, after their immunophenotypic analysis.

#### 2.2.3. Wharton Jelly (WJ)-Derived Mesenchymal Stem Cells (WJ-hMSCs)

Human Umbilical Cords (*n* = 4) were obtained from full-term cesarean deliveries with informed consent. The umbilical vessels were removed after washing with 1X PBS. Within 1 to 2 h after receiving the cords, the mesenchymal tissues (Wharton jelly) were scraped from the cords, minced into approximately 5-mm^2^-small fragments, and plated onto culture dishes in the presence of a proliferative medium composed of α-minimum essential medium (α-MEM) with 15% human serum (Biopredic International, Rennes, France), 2 mmol/L L-glutamine, and 0.1 mg/mL kanamycin, and incubated in a CO_2_ incubator. The media were changed after every 3 days to allow the cells to migrate out from the tissue fragments. WJ-hMSCs were expanded at the ratio of 1:2 until they were 90–95% confluent and were subcultured three times and finally subjected to induction for exosomes’ collection.

### 2.3. Trilineage Differentiation Potential of hMSCs

Human MSCs isolated from BM, AD, and WJ were characterized for their multi-differentiation potential. The hMSCs were differentiated into osteogenic, chondrogenic, and adipogenic lineages, as per the previously established laboratory protocols, as discussed briefly below [24]. After their differentiation, adipogenic, osteogenic, and chondrogenic differentiation experiments were performed using standard protocols for human mesenchymal stem cell functional identification.

For adipogenic differentiation, hMSCs were seeded at a density of 3.7 × 10^4^ cells/well and maintained in a culture medium until 100% confluence. At the third passage, cells were then exposed to an adipogenic differentiation medium containing DMEM-LG (Gibco, Invitrogen, Waltham, MA, USA), indomethacin (100 µM), dexamethasone (1 µmol/L), 3-isobutyl-1-methylxanthine (500 µM), insulin (1 µg/mL) (Sigma, St. Louis, MO, USA), and 10% FBS (Hyclone, Logan, UT, USA) for 3 weeks. The media were changed after every 3 days. Lipid droplets of the resultant-differentiated cells were confirmed using Oil red staining with uninduced cells and were used as experimental control cells (Sigma-Aldrich, St. Louis, MO, USA).

For osteogenic differentiation, 4.2 × 10^3^ hMSCs were seeded per well. After the cells reached 50–70% confluence, the medium was replaced with an osteogenic differentiation medium consisting of DMEM-LG, ascorbic acid-2-phosphate (50 µM), dexamethasone (0.1 µM), ß-glycerophosphate (10 mM) (Sigma, St. Louis, MO, USA), and 10% FBS and kept for 4 weeks with intermittent change in media after every 2 days. To assess osteogenic differentiation, Alizarin Red S staining (Sigma-Aldrich, St. Louis, MO, USA) was performed to detect the calcium-rich extracellular matrix.

For chondrogenic differentiation, hMSCs were differentiated into the chondrogenic lineage using a commercially available kit (Gibco, Invitrogen, Waltham, MA, USA). Briefly, the hMSCs were detached from the monolayer using TrypLE Express at the third passage and resuspended at a density of 1.6 × 10^7^ viable cells/mL. Then, 5 μL of the cell suspension were seeded in a 35-mm cell culture dish to generate a micromass culture. The media were refed after every 3 days until 14 days before termination. Uninduced hMSCs were used as experimental controls, and the differentiation was confirmed with Alcian Blue staining (Sigma-Aldrich, St. Louis, MO, USA).

### 2.4. Isolation of Exosomes from hMSCs

The serum-free conditioned media from hMSCs from different donors for all tissue-specific sources cultured for 48 h were collected individually and processed for exosome isolation, as previously described [23]. The cellular debris was briefly removed by centrifugation at 300× *g* for 10 min, followed by centrifugation at 10,000× *g* for 30 min to remove microvesicles. The conditioned media were loaded slowly over 30% sucrose solution, forming a layer and ultracentrifuged at 100,000× *g* at 4 °C for 90 min using Sorvall™ WX 90+ ultracentrifuge in a swinging bucket rotor (Thermo Scientific, Waltham, MA, USA). The supernatant was discarded, and the sucrose layer was resuspended in 1X PBS and ultracentrifuged again at 100,000× *g* at 4 °C for 90 min to pellet down the exosomes. Finally, the pellet was resuspended in 500 µL of filtered 1X PBS. These exosomes obtained from different donors for all tissue-specific MSCs were aliquoted and stored at −80 °C for further use.

### 2.5. Nanoparticle Tracking Analysis (NTA)

The exosomes were diluted (1:10) in 1X PBS for NTA by NanoSight LM20 (Malvern Instruments Company, NanoSight, Malvern, UK). The Brownian motion of each particle was tracked between frames, and the size was calculated using the Strokes–Einstein equation.

### 2.6. Transmission Electron Microscopy (TEM)

The exosomes’ suspension was diluted at a concentration of 1:1000 in PBS and was placed on Formvar/carbon-coated copper grids and allowed to adsorb for 5 min in a dry environment. The grids were washed in drops of 1X PBS and stained with 2% phosphotungstic acid solution for 1 min. After that, grids were air-dried and were observed under the electron microscope (Tecnai, FEI, Hillsboro, OR, USA).

### 2.7. Western Blotting

To characterize the presence of CD63 and Alix exosomal proteins, exosomes were lysed in RIPA buffer containing 1 mM PMSF and protease inhibitors. The protein concentration of lysates was determined by using the Bicinchoninic Acid Assay (BCA) protein assay kit (Pierce, Waltham, MA, USA). Then, 20 μg of protein were subjected to 12.5% SDS-PAGE under non-reducing conditions for CD63 and under reducing requirements for Alix antibodies. The gel was transferred to a PVDF membrane (MDI) using a wet transfer system (Bio-Rad, Hercules, CA, USA). The blot was blocked in 5% non-fat skimmed milk in 1×TBS-T solution followed by incubation in CD63 primary antibody (1:5000 dilution; Abcam, Cambridge, MA, USA) and Alix primary antibody (1:1000; Santa Cruz, Dallas, TX, USA) overnight at 4 °C. The blot was then washed and incubated with an HRP-conjugated secondary antibody (1:10,000; Invitrogen, Waltham, MA, USA) and developed using an ECL imager (Invitrogen, Waltham, MA, USA).

### 2.8. Small RNA Sequencing of RNA Isolated from hMSCs and Their Respective Exosomes

Total RNA isolation from exosomes was done using miRCURY Exosome RNA Isolation kit (Exiqon, Vedbaek, Denmark) according to the manufacturer’s protocol. Small RNA sequencing was done using Illumina Hi Seq 2500 (NIBMG, Kolkata, India).

### 2.9. Library Preparation, Sequencing, and Data Analysis

Library preparation, sequencing, data analysis, and RNA quality was assessed using an Agilent Tape Station (Agilent, Palo Alto, CA, USA), and RNA concentration was quantified by Qubit 4.0 spectrophotometer (Life Technologies Holdings Pte Ltd., Mapletree, Singapore). The library was prepared for sequencing using the small RNA library preparation kit according to the manufacturer’s protocol (Illumina, San Diego, CA, USA). The resulting libraries were quantified using the KAPA Library Quantification kit (KAPA Biosystems, Wilmington, MA, USA), and quality and size were checked using the Agilent High Sensitivity DNA Kit. Small RNA-seq libraries were sequenced using single directional end sequencing chemistry on Nova-seq instruments by following the manufacturer’s protocols (Illumina, San Diego, CA, USA). The raw reads corresponding to Illumina RNA-seq FASTQ files were aligned in Strand NGS software v3.4 against the human reference genome (Build hg19, Gene feature: UCSC Genes, Library layout: single-end). Quantification was done to obtain the read densities by Strand NGS software. Differential expression analyses of miRNA from different groups were performed using the DESeq script two via the script editor in Strand NGS with a *p*-value of ≤0.05 and fold change 1.5 or 2 fold. Pathway analysis was performed using Ingenuity Pathway Analysis (IPA) software (Qiagen Inc., Venlo, The Netherlands, https://www.qiagenbioinformatics.com/products/ingenuitypathway-anlysis, accessed on 20 April 2020) by loading 1.5- or 2.0-fold statistically significant miRNA list (0.05 *p*-values).

### 2.10. MicroRNA Profiling

The top miRNAs identified from the small RNA sequencing data were validated in the hMSCs isolated from BM, AD, and WJ and their respective exosomes. Briefly, total RNA was isolated from hMSCs from different sources and their exosomes. Ten ng/mL of total RNA were converted to cDNA using MystiCq microRNA cDNA Synthesis kit (Sigma Aldrich, St. Louis, MO, USA). Ten µL of cDNA solution were diluted 10 times with nuclease-free water. From this solution, 2 µL of diluted cDNA were used for the qRT-PCR using the TB Green Premix Ex Taq II (Tli RNase H Plus). The plate was run on a Quanta-Studio7 (Applied Biosystems, Waltham, MA, USA) real-time detection system for 40 cycles. Data were analyzed by using the Ct values, and cellular miRNAs were normalized against U6 while miRNAs from exosomes were normalized against highly enriched miRNA, i.e., miR-16.

### 2.11. Neural Stem Cells C17.2 Culture

Mouse-derived multipotent neural stem cells C17.2 were obtained from Dr. Anirban Basu, Laboratory of Virology, National Brain Research Centre, India. The cells were routinely cultured to 70–80% confluency in DMEM medium supplemented with 10% FBS, 5% Horse serum (HS), and 1% PS + 1% glutamine, and cultured in a standard, humidified incubator at 37 °C with 5% CO_2_. Cells were seeded in six-well plates for differentiation experiments at a seeding density of 0.5 × 10^6^ incomplete DMEM, which was changed to differentiation medium 1 day post-seeding. Cells were incubated with the differentiation medium consisting of DMEM:F12 medium (Gibco, Waltham, MA, USA) with N2 supplements, Nerve growth factor (NGF), and Brain-Derived Neurotrophic Factor (BDNF) (10 ng/mL).

### 2.12. The qRT-PCR Assay

For studying the effect of exosomes from different hMSC sources on neurogenesis, exosomes equivalent to 30 μg/mL of protein as measured by BCA were added along with the differentiation media and incubated for 7 days with an intermittent change of media after every 3 days. After 7 days, total RNA from the cells was isolated using a RNeasy kit (Qiagen, Venlo, The Netherlands) with in-column DNase digestion. Then, 500 ng of total RNA were reverse-transcribed using random hexamers and ImProm-II reverse transcription system (Promega, Madison, WI, USA). All qPCR reactions were performed in a 5-µL mixture containing 2× Fast SYBR-Green mixes (Invitrogen, Waltham, MA, USA), 2.5 µL diluted cDNA, and 0.5 µL primer. For all experiments, GAPDH levels were used for normalization. The primers synthesized by Qiagen, Venlo, The Netherlands, were used for quantifying the various gene transcripts to study the markers for neurogenesis in C17.2 cells. The primers used were GAPDH (F) 5′CCTGCCAAGTATGATGAC; (R) 3′ GGAGTTGCTGTTGAAGTC); Nestin (F) 5′CTCAACCCTCACCACTCTATTT3; (R) 5′CTGTGGCTGCTTCTTTCTTTAC3; GFAP (F) 5′AATGCTGGCTTCAAGGAGAC3; (R) 5′AAGCGGACCTTCATGTA3); MAP 2 (F) CTCTGCCTCTAGCAGCCGAA; (R) CACCACTTGCTGCTTCCTCC and OLIG2 (F) GGCGGTGGCTTCAAGTCATC; and (R) TAGTTTCGCGCCAGCAGCAG.

### 2.13. PBMCs’ Isolation and Proliferation

PBMCs were isolated from buffy coats using density gradient centrifugation (Lymphoprep, StemCell Technologies, Vancouver, BC, Canada) and were resuspended in RPMI with 10% FBS and stained with CFSE dye (CFSE; Molecular Probes, Eugene, OR, USA). After that, 30 μg of exosomes were incubated with 0.1 × 10^6^ CFSE-labeled cells in a round-bottom, 96-well plate. To activate T-cell proliferation, ImmunoCult™ Human CD3/CD28 T Cell activator was used at a concentration of 25 µg/mL. T-cell proliferation was analyzed after 6 days using flow cytometry BD FACS Verse (BD Biosciences, Franklin Lakes, NJ, USA), and data were analyzed using the FlowJo Vx.0.7 (FlowJo, LLC, Ashland, OR, USA).

### 2.14. Neutrophil Isolation

Neutrophils were isolated using an EasySep Direct Human Neutrophil isolation kit (StemCell Technologies, Vancouver, BC, Canada) according to the manufacturer’s protocol. Briefly, blood was withdrawn from healthy donors with prior consent in EDTA vials and the antibody cocktail was added to it (50 μL/mL of blood) for 5 min. After that, rapidsphere beads (50 μL/mL of blood) were added and incubated for 5 min. The suspension was diluted with Ca^2+^/Mg^2+^ free PBS and kept in the magnetic stand for 5 min. The suspension was carefully decanted in another tube in one continuous flow. The same amount of rapidsphere beads was added to the suspension for 5 min and then kept on the magnetic stand again for 5 min. This step was repeated again but without the addition of rapidsphere beads until a clear suspension was obtained. Finally, the pure neutrophil suspension was centrifuged at 800 rpm for 5 min at RT. The cells were then resuspended into RPMI-1640 media (HiMedia, Mumbai, India) and were counted using a hemocytometer before downstream applications.

### 2.15. Neutrophils’ Phagocytosis Assay

Neutrophils were treated with exosomes at a concentration equivalent to 30 µg/mL protein. After 8 h of treatment, the phagocytic capacity was examined. Briefly, a suspension of CFSE-labeled *E. coli* bacteria was prepared and was resuspended at a density of 10^8^ particles per mL in RPMI-1640 medium. Then, 10 μL of this suspension were added to neutrophils, and the cells were allowed to phagocytose the bacteria at 37 °C in a water bath for 60 min. After incubation, the cells were washed with PBS four times to remove any extracellular bacteria and were transferred to FACS tubes for data acquisition. The data were analyzed on FlowJo software and the percent of CFSE-positive cells was plotted in the graph. 

### 2.16. Apoptosis Assay

To examine the effect of exosomes isolated from hMSCs on the apoptosis of neutrophils, the cells were incubated with exosomes at a 30 µg/mL concentration for 24 h. The cells were washed with 1X PBS post-incubation and resuspended into the annexin binding buffer. The cells were then stained with annexin V-FITC antibody and propidium iodide as per the manufacturer’s protocol (Apoptosis Detection Kit; Abgenex, Bhubaneswar, India) and incubated for 20 min. The cells were acquired using FACS and analyzed by using FlowJo software.

### 2.17. Visualization of NETs and Measurement of Extracellular DNA Release

The effect of exosomes isolated from hMSCs on NET induction and the release of extracellular DNA was studied by seeding neutrophils at a density of 0.2 × 10^5^ cells per well on a coverslip and 96-well plate, respectively, and incubating them with exosomes (30 µg/mL) at 37 °C with 5% CO_2_ for 6 h. For NET visualization, the cells were fixed with 4% PFA and then permeabilized with 0.1% TritonX-100. The cells were blocked with 5% BSA and stained with MPO, Citrullinated histone antibodies, and DAPI stain. The coverslips were visualized under a Leica SP8 confocal microscope. For studying the release of extracellular DNA, the membrane-impermeable, DNA-binding dye Sytox Green (0.25 μm) (Invitrogen, Waltham, MA, USA) was added to bind extracellular DNA for 15 min. The fluorescence intensity was measured at Excitation/Emission of 504/523 nm using a spectrophotometer (Thermo Fisher Scientific, Waltham, MA, USA).

### 2.18. Angiogenesis Study

The effect of exosomes on angiogenesis in endothelial cells was studied using HUVEC cells isolated from the blood of healthy donors. The endothelial cells were seeded on a matrigel at a seeding density of 0.5 × 10^5^ cells per well in six-well plates and incubated with exosomes for 24 h. Tube formation assays were performed according to the manufacturer’s protocol of an Angiogenesis Starter Kit (Life Technologies, Carlsbad, CA, USA). Structures were observed under a bright microscope.

### 2.19. Statistical Analysis

Data were analyzed using GraphPad PRISM 8.0 (San Diego, CA, USA) and presented as mean ± SD/SE of at least three independent experiments with three replicates. Statistical significance was determined by using either Brown Forsythe and Welch ANOVA or Two-way ANOVA followed by an appropriate post hoc test wherever applicable, and *p* < 0.05 was considered statistically significant.

### 2.20. Data Availability

The data discussed in this publication are deposited in NCBI’s Gene Expression Omnibus [25] and are accessible through GEO series accession number GSE153752.

## 3. Results

### 3.1. Characterization of hMSCs and Their Exosomes

The cryopreserved hMSCs isolated from bone marrow, Wharton jelly, and adipose tissue of adult donors (18–50 years old) were revived and cultured for expansion (Appendix A). To validate their candidacy, subsequent marker studies were performed. The hMSCs were visualized using Phase-contrast microscopy, which revealed the spindle-shaped morphology of tissue-specific hMSCs (Figure 1A). These hMSCs showed a tri-lineage differentiation potential as evaluated by staining with Alizarin red, Oil red O, and Alcian blue for Osteocytes’, Adipocytes’, and Chondrocytes’ differentiation, respectively (Figure 1B). These hMSCs were also characterized for their surface marker profiling by flow cytometry and were found positive for CD29, CD73, CD90, CD105, and HLA I and negative for HLA II and hematological markers CD34/45 (Appendix A).

The hMSCs from these three different sources were cultured, and the cell culture supernatant was collected from passage 3–5 cells for the purification of exosomes using ultracentrifugation. Transmission Electron Microscopy (TEM) revealed their morphology, and they were found to be both round and cup-shaped vesicles, corresponding to the size of exosomes and maintaining their integrity (Figure 1C). Nanoparticle tracking analysis (NTA) of purified exosomes revealed that hMSCs-derived exosomes possess a canonical diameter size distribution, with a mean diameter of 126 nm (*n* = 3 donors) with a peak < 200 nm (Figure 1D). Western blots for CD63 and Alix also confirmed that the isolated vesicles were exosomes (Figure 1E).

### 3.2. MicroRNAs’ Profiling of hMSCs and Their Exosomes

To gain insights into microRNAs detected in hMSCs and their exosomes, we performed a transcriptomic study to reveal the miRNAs in hMSCs and exosomes. Total RNA isolated from parental cells and exosomes was subjected to next-generation sequencing followed by data analysis. To define the small RNA composition, we degraded the unprotected RNA during exosomes’ preparation by adding exogenous RNase A. Subsequently, cellular and exosomal RNA were subjected to a bioanalyzer to study the small RNA profile of exosomes. The bioanalyzer revealed a characteristic peak between 20 and 70 nucleotides of miRNAs and tRNAs in both samples. Firstly, we analyzed small RNA, rRNA, and other RNA categories within the cells and exosomes. We considered all the miRNAs detectable in at least two samples and had more than 10 reads.

The total miRNAs detected in hMSCs from AD, BM, and WJ were 279, 221, and 275, respectively (Figure 2A). A total of 204 miRNAs were found to be shared between hMSCs from all the sources. In comparison to hMSCs, exosomes derived from them contained a lesser number of miRNAs. A total no. of 122 miRNAs was found common between hMSCs and exosomes (Figure 2A). Totals of 96, 92, and 108 miRNAs were detected in AD-, BM-, and WJ-derived exosomes. Interestingly, 81 miRNAs were common in the exosomes irrespective of their sources (Figure 2A). The expression levels of these 81 miRNAs in the exosomes isolated from the three sources are represented in the form of a heatmap (Figure 2B). The expression levels of seven common miRNAs, as observed in the transcriptomic data in cells and exosomes, are represented in Figure 3A.

We further validated the expression pattern of miRNAs such as miR-145-5p, 181c-5p, 486, 143-5p, 149-5p, 146a-5p, 125b-5p, and 132-5p in RNA isolated from hMSCs (Figure 2C) and exosomes (Figure 3B), respectively, using RT-PCR. These miRNAs were taken based on the IPA analysis as well as their functions reported to have various roles in promoting autophagy and T cell activation [22], inhibiting apoptosis [26], and acting as a control switch between angiogenesis and cell death [27] (Appendix A). The BM-hMSCs had increased expression of miR-145-5p and 143-5p, while WJ-hMSCs expressed miR-149-5p, 146a-5p, and 125b-5p the most as compared to other hMSCs. The expression of miRNA-181c and 486 was increased in both BM-hMSCs and WJ-hMSCs, while miRNA-132-5p was increased in AD-hMSCs as well as in BM-hMSCs (Figure 2C). The expression levels of all the miRNAs were relatively lower in AD-hMSCs as compared to BM-hMSCs and WJ-hMSCs except for miR-132-5p (Figure 2C). In the case of the expression of these miRNAs in exosomes, BM-derived exosomes expressed miR-125b-5p, 181c-5p, 149-4p, and 143-5p abundantly as compared to exosomes from other sources, while WJ-derived exosomes highly expressed miRNA-146a-5p, 132-5p, and 145-5p (Figure 3B). The miR-486 was highly expressed in both BM- and WJ-derived exosomes. As observed with hMSCs, AD-derived exosomes showed the least expression of most of the miRNAs. Surprisingly, the expression pattern of these miRNAs was not similar in parental hMSCs and their respective exosomes.

To determine the potential targets of these miRNAs, IPA analysis was performed. According to the analysis, these miRNAs targeted pathways such as cellular development, HOTAIR regulatory pathway, cancer drug resistance pathway, TH1, and TH2 activation pathway, and apoptosis pathway as well (Figure 2D).

### 3.3. Effect of hMSCs-Exosomes on Neurogenesis

The hMSC-derived exosomes contained many miRNAs (miR-125 [28,29], miR-145 [30], miR-18 [31,32], miR-21 [33]) abundantly associated with brain functions. Therefore, the potential of these exosomes on the differentiation of multipotent neural stem cells (C17.2) cells was studied. The exosomes were incubated with C17.2 cells in the presence of differentiation media for 7 days. The expression of differentiation markers such as Nestin, GFAP, MAP2, and OLIG2 was measured after 7 days for studying neural differentiation using RT PCR [34]. The exosomes from all the sources showed enhanced neurogenesis as evident by the expression profile of the different markers. The mRNA expression level of the neural progenitor marker Nestin was enhanced in exosomes from all the sources, indicating the augmented stemness and less differentiation of C17.2 cells in the presence of media and exosomes. Only BM-derived exosomes significantly increased the expression of GFAP, which pointed towards the increased ability for astrocyte differentiation by BM-derived exosomes. An additive effect of exosomes from BM and WJ on the expression of MAP2 was observed, while OLIG2 was significantly upregulated only in the case of BM-derived exosomes, indicating that the exosomes accelerated the differentiation process in the presence of differentiation media (Figure 4A).

It was already reported that miR-145 could regulate neural stem cell differentiation [35]. The hMSC-derived exosomes contained miR-145; so, it might be possible that the additive effect of exosomes on the neurogenesis process may occur due to miR-145 in the exosomes. Therefore, we tested the effect of miR-145 overexpression and inhibition using miR-145 mimic and inhibitor, respectively, to C17.2 cells in the presence of differentiation media. An increase in expression of Nestin was observed in cells transfected either with the mimic or with the inhibitor for unknown reasons. The expression of Nestin was downregulated in the presence of differentiation media when compared to the undifferentiated C17.2 progenitor cells (Data not shown). The transfection of miR-145 mimic along with DM significantly augmented Nestin, GFAP, MAP2, and OLIG2 mRNA expression, suggesting increased differentiation by the overexpression of miR-145. On the other hand, miR-145 inhibition had no effect on the expression of Nestin and OLIG2. However, GFAP and MAP2 were found to be significantly attenuated in the presence of miR-145 inhibitor as compared to mimic-145 transfecting cells.

### 3.4. Effect of hMSC-Derived Exosomes on the Expansion of T Cells In Vitro

Our in silico analysis also suggested that hMSCs-exosomes contain miRNAs that can modulate the TH1 and TH2 activation pathways. From our IPA study, we found that mir-146, mir-155, mir-21, and mir-29 could be associated with TH1 and TH2 activation pathways. However, it is not known how these exosomes from different sources behave in response to immune regulation. Therefore, the immunomodulatory capacities of hMSCs-exosomes were investigated using T-cell proliferation assays. The effect of hMSCs-derived exosomes was studied on the expansion of PBMCs stimulated with anti-CD3/CD28 antibodies. Our study suggested that exosomes from all different sources can suppress T cell proliferation (Figure 5A). However, BM- and WJ-derived exosomes have a better suppressive capacity than AD-exosomes (Figure 5B).

### 3.5. The Effects of hMSCs-Exosomes on Neutrophil Phagocytosis

The phagocytic ability of neutrophils is an essential tool to engulf and eliminate pathogens from circulation. To assess the effect of hMSCs-exosomes on the phagocytic capacity of neutrophils, the cells were treated with exosomes for 24 h and were then allowed to phagocytose CFSE-labeled *E. coli*. The results show that hMSCs-derived exosomes could significantly increase the phagocytic capacity of neutrophils depending upon the source of exosomes (Figure 6A). Treatment with AD-exosomes enhanced the phagocytic ability of neutrophils more than the treatment with exosomes from BM and WJ, as observed by the percentage of cells phagocytosed by the neutrophils (Figure 6A). A report suggests that miR-378a can modulate macrophage phagocytosis [36]. Overexpression of miR-378 promoted phagocytosis. The level of miR-378 was less in BM-Exo-treated cells as compared to AD-Exo and WJ-Exo, (Figure 6E, Left panel) and less phagocytosis activity was observed in neutrophils treated with BM exosomes.

### 3.6. The Effects of hMSC-Exosomes on Survival and NET Formation of Neutrophils

We designed this study to investigate how hMSC-exosomes affect the apoptosis and function of neutrophils isolated from different tissue sources using Annexin V and PI staining and NET formation studies, respectively. After 24 h of treatment with exosomes from three various sources, it was found that after 24 h post-incubation, neutrophils were mostly Annexin V-positive cells, suggesting they are early apoptotic cells. A much smaller percentage of cells was both Annexin V- and PI-positive (Appendix A). The early apoptosis of neutrophils was significantly delayed by hMSC-exosomes (Figure 6B). Especially, Neutrophils incubated with BM-exosomes showed significant delay, whereas a statistically non-significant but consistent delay was observed in neutrophils incubated with AD and WJ compared with untreated. We also checked miR-486 expression upon 24 h post-treatment with exosomes. The miR-486 has been reported to promote proliferation and suppress apoptosis in myeloid cells [36]. In our study, we observed a decreased miR-486 level and increased apoptosis in WJ-treated samples as compared to BM-exosome-treated neutrophils (Figure 6E, Right panel).

Another vital aspect of neutrophils’ activation in context to bacterial and viral infection is to release Neutrophil extracellular traps (NETs), which will restrict the spread of the pathogens. Our study observed that incubation of freshly isolated neutrophils with AD-exosomes and WJ-exosomes induced NETs more efficiently than exosomes isolated from BM (Figure 6D). Many delobulated or defused nuclei can be seen in neutrophils treated with AD and WJ exosomes, but an intact nucleus (relatively smaller size) was observed in control PBS-treated neutrophils (Appendix A). The release of extracellular DNA from neutrophils was studied by measuring the fluorescence of Sytox orange (Figure 6C). Further, we observed increased miR-378a expression upon treatment with AD and WJ exosomes (Figure 6E, Left panel). Increased miR-378a is associated with increased NET formation [20], which further correlates with our imaging data.

### 3.7. Effects of hMSCs-Exosomes on Angiogenesis

Exosomes from hMSCs have been shown to exhibit angiogenesis-promoting characteristics. Several microRNAs (miRNAs) have been identified that play a specialized role in promoting angiogenesis. These comprise miR-132, miR-145, miR-125, and other related microRNAs [21]. As a result, according to our bioinformatics’ analysis, some of these well-known miRNAs were found in exosomes derived from different tissue-specific MSCs. However, to validate their functional role, we performed tube formation assay in which the endothelial cells (HUVEC) were incubated with the exosomes from three different sources at a concentration of 30 µg/mL protein equivalent exosomes for 24 h. The subsequent day angiogenesis assay was performed, and, finally, the images were captured (Figure 7A). The total no. of loops, branches, and junctions was counted, and the length of branches was also calculated. According to the findings, all tissue-specific MSC exosomes were able to considerably increase tube formation when compared to the control group. A striking difference between exosomes from BM and WJ was the presence of a considerable increase in loop formation, junctions, and branching when compared to exosomes from AD. (Figure 7B–D). The longest branches were observed in WJ-exosomes-treated cells (Figure 7E). Because of this, WJ- and BM-derived exosomes were found to significantly accelerate the process of in vitro angiogenesis and can likely play a more important part in regeneration. Furthermore, these findings were consistent with the miRNA data (Figure 3), which showed that miR-132 and miR-145 were found to be increased in WJ-MSCs (Figure 3), while miR-125 was observed to be upregulated in BM-MSC-derived exosomes.

## 4. Discussion

MSCs-based therapies have a huge potential in the field of tissue repair [37] and regenerative medicine [38,39]. Studies have shown that hMSCs are beneficial in wound healing when used in several stages of tissue repair and regeneration in both the disease models and clinical evaluations [39]. For example, hMSCs are reported to be clinically efficient in scar treatment [40,41], wound healing [39,40,42], hair loss treatments [43,44,45], and breast augmentation [46,47]. Interestingly, recent studies also suggested that these MSCs can also be used to treat COVID-19-induced pneumonia [48,49]. Still, they are associated with several inherent risks like occlusion in the microvasculature, transformation of the transplanted cell into inappropriate cell types, or cancer and pro-arrhythmic side effects [50,51]. Recently, a new concept of cell-free therapy has been developed where hMSCs-derived small extracellular vesicles (exosomes) have been utilized. In contrast with large living cells, nano-sized exosomes would not occlude microvasculature, transform into inappropriate cell types, or persist as permanent grafts upon cessation of therapies. The most attractive feature of exosomes is their ability to mimic the effect of the mother cells. The ability of exosomes to mimic the effects of mother cells can be attributed to their cargoes consisting of several miRNAs, proteins, and lipids. In this study, we compared the hMSCs’ and exosomal miRNAs’ cargo between AD-, BM-, and WJ-derived hMSCs to understand their miRNA landscape. Small RNA sequencing of the hMSCs and their exosomes and target prediction analysis were performed to get deeper molecular insights into their cargoes. Data integration highlighted comprehensive and comparative information about the common and uniquely present miRNAs in these exosomes. We used network analyses to comprehensively understand the main pathways and biological processes targeted by the abundant miRNAs present in the exosomes isolated from different hMSCs. Through Ingenuity Pathway analysis, we identified the HOTAIR regulatory pathway, the adipogenesis pathway, vascular and angiogenesis processes proliferation, TH1 and TH2 activation pathway, and apoptosis. Several microRNAs, including miR-146, miR-10, miR-322, miR-378, miR-181, miR-143, miR-486, let-7 family, miR-155, and miR-148 that were found in the exosomes, could be associated with these pathways. Our study also confirmed that there is variability in the abundance of some miRNAs within different sources in the parental cells (e.g., miR-486-5p, miR-191-5p, and miR-146a-5p) and in exosomes as well (e.g., miR-145, miR-125b, miR-486, miR-181, miR-149, and miR-143). The expression pattern of these miRNAs in exosomes was different than that of their parental hMSCs. Except for a few miRNAs (miR-143-5p, 181c-5p in BM, and miR-146a-5p in WJ) the relative expression level of most miRNAs in exosomes was drastically reduced as compared to their source. It is important to note that the methods of reporting the expression of cellular miRNA and exosomal miRNAs are different. For cellular miRNAs, U6 was used to normalize the miRNAs’ expression. However, for exosomal miRNAs’ expression, we used miR-16 expression to normalize the data, as described earlier [52]. Depending on abovementioned reference, different miRNAs were used for normalization; therefore, the expression levels of miRNAs from exosomes and hMSCs may vary.

Apart from this, several miRNAs were abundant in exosomes associated with brain cells’ functions, suggesting that hMSCs-exosomes could be a valuable tool to treat brain-related diseases [53,54]. Exosomes from different sources accelerated the differentiation of neural cells, as evident by increased expression of GFAP, MAP2, and OLIG2. In contrast, the expression of Nestin was not attenuated in the presence of these exosomes. Nestin is a type VI intermediate filament protein, characterized as a marker of undifferentiated neural stem cells, which vanishes upon cell differentiation [34]. Compared to the untreated progenitor cells, the expression of Nestin showed decreased expression in the presence of DM. Still, its expression increased in the presence of exosomes for unknown reasons. BM-derived exosomes significantly increased the expression of GFAP, MAP-2, and OLIG2. GFAP, a glial fibrillary acidic protein, is an intermediate filament (IF) protein and a well-known marker for astrocytes [34]. The OLIG2 is one of the first factors expressed in neural progenitors during early neurogenesis [34]. MAP-2 is a neuron-specific cytoskeleton protein that is used as a marker of neuronal phenotype [34]. Apart from BM exosomes, WJ-derived exosomes showed a significant increase in the expression of GFAP only. It has been reported that MSCs from WJ and BM release several factors that promote neurogenesis, but WJ-MSCs expressed more genes associated with neurogenesis [55]. Exosomes from MSCs also promoted neurogenesis in animal models of Alzheimer disease [54] and traumatic brain injury [56] and can enhance recovery in the case of spinal cord injury [57].

Apart from neurogenesis, the exosomes also promoted angiogenesis in HUVEC cells. We reported that exosomes from BM and WJ showed better angiogenic properties than AD. These results were in line with the previously published reports where WJ and BM have been shown to exhibit enhanced angiogenetic potential [55,58]. However, there are reports that AD-MSCs can promote angiogenesis by increasing endothelial cell differentiation and cell migration [59], while BM-MSCs express VEGF that promotes angiogenesis [60]. Exosomes are reported to contribute to angiogenesis through suppressing the expression of factor-inhibiting HIF-1, promoting hypoxic signaling, or upregulating the expression of proangiogenic factors [61]. Several microRNAs (miRNAs) have been identified that play a specialized role in promoting angiogenesis. This comprises miR-132, miR-145, miR-125, and other related microRNAs, which are also present in detectable amount in the exosomes. In fact, miR-132 and miR-145 were found to be increased in WJ-MSCs (Figure 3), while miR-125 was observed to be upregulated in BM-MSC-derived exosomes. Therefore, our findings can contribute to a better understanding of the molecular mechanisms underlying the proangiogenic responses induced by hMSCs, suggesting a key regulatory role for microRNAs delivered by Exosomes.

Apart from their neurogenic and angiogenic properties, exosomes also showed immunomodulatory properties in PBMCs and neutrophils. We also observed the cell type-specific effect of exosomes in apoptosis and T cell proliferation. These functions probably correlate with the abundance of miRNAs in exosomes. Exosomes from all the sources inhibited the proliferation of PBMC proliferation. This study further confirms that exosomes retain immunosuppressive abilities on the expansion of PBMC, similar to their parental cells [62]. Several miRNAs, including miR-146, miR-155, and let-7e detected in exosomes derived from hMSCs, are known to have immunomodulatory roles [63,64,65]. The effect of exosomes on apoptosis and NET formation in neutrophils was also studied. Neutrophils are the significant population of innate immune cells with multiple proteases, antimicrobial peptides, and reactive oxygen species (ROS) for killing pathogens. The incubation of neutrophils with BM and AD-exosomes improved neutrophil lifespan as compared to WJ-exosomes. BM-exosomes showed the most significant delay in apoptosis and no effective results were found in the case of exosomes from WJ. Still, WJ- and AD-derived exosomes have been reported to improve the survival of neutrophils [66,67]. AD exosomes can even delay the apoptosis of neutrophils isolated from severe congenital neutropenia patients in in vitro conditions [67]. There are several other diseases where neutropenia is a signature manifestation of pathogenesis. Therefore, effective strategies for improving the function and lifespan of the existing neutrophils in these patients are necessary. Additionally, exosomes from AD and WJ were found to have a higher capacity to release extracellular DNA from neutrophils. NETosis has both beneficial as well as detrimental effects on the host. The extracellular traps are described to be formed in vivo and contribute to the clearance of infections [68,69]. Treatment of neutrophils with exosomes from AD increased their phagocytic capacity, while BM-exosomes did not affect the same. WJ-derived exosomes also showed insignificant increase, and they are reported to increase the phagocytic capacity in neutrophils [66]. Moreover, expression of miR-378a, which are associated with increased phagocytosis [36] and NET formation, correlated with their expression in exosome-treated cells. However, whether single miRNA or multiple miRNAs in combination can modulate a cell’s phenotype and function needs further investigations. Therefore, the effect of these EVs on neutrophils in context to inflammation and infection needs to be evaluated in future studies.

## 5. Conclusions

Our data provide the MSC miRNA landscape with a more in-depth insight into the differences and similarities between hMSCs from different origins and predict their biological functions on a system level. The miRNAs identified were known to play a role in immune modulation, neurogenesis, regulating apoptosis, and T cell proliferation. Exosomes from BM-hMSCs are more efficient in promoting neuronal differentiation and delaying neutrophil apoptosis and PBMC proliferation compared to AD-hMSCs and WJ-hMSCs. On the other side, treatment with exosomes from AD-hMSCs increases phagocytic capacity and NET formation in neutrophils. Angiogenesis is highly induced by WJ-hMSCs exosome treatment. Thus, our study provides a resource for further improving our understanding of exosome cargoes and their interaction with immune cells.

## Figures and Tables

**Figure 1 biomedicines-10-00069-f001:**
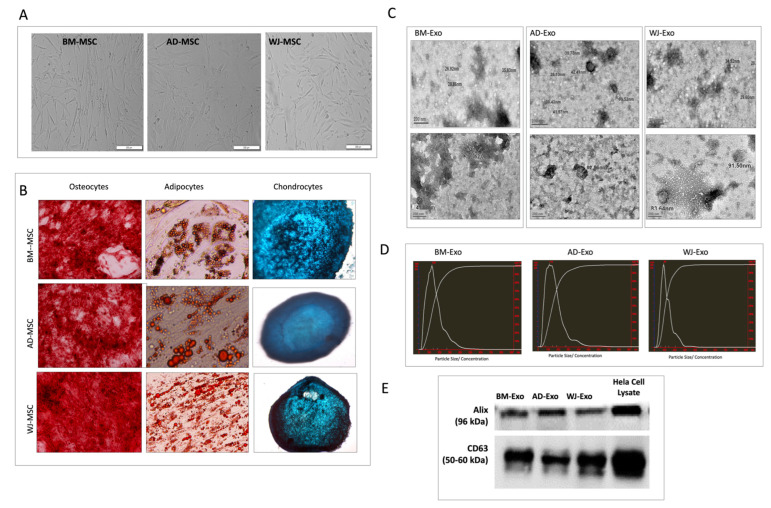
Characterization of hMSCs and their exosomes. (**A**) Phase-contrast microscopy of tissue-specific hMSCs (magnification 10×) (upper panel). (**B**) Trilineage staining of hMSCs: Alizarin Red staining for osteocytes’ differentiation, oil red O staining for Adipocytes’ differentiation, and alcian blue staining for chondrocytes’ differentiation (magnification 20×) (lower panel). (**C**) Morphology analysis of exosomes by Transmission Electron Microscope (TEM) (Bar = 200 nm; 19 KX) (right panel). (**D**) Exosomal size distribution analysis using nanoparticle tracking analysis (NTA). (**E**) Western blots showing expression of exosomal surface marker CD63 and ALIX.

**Figure 2 biomedicines-10-00069-f002:**
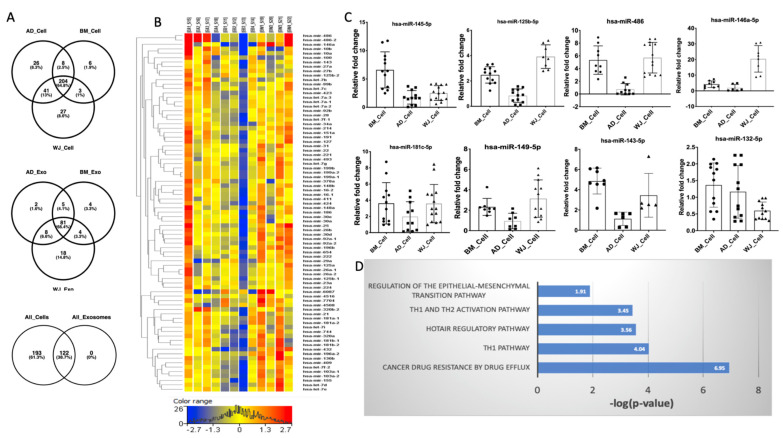
The miRNAs’ profiling of hMSCs and their exosomes. (**A**) Total number of miRNAs detected and commonly present in hMSCs and exosomes. (**B**) Heat map of a total of 81 miRNAs expressed in all three different sources of exosomes. EA = Exosomes Adipose tissue, EB = Exosomes Bone marrow, EW = Exosome Wharton Jelly; 1, 2, 3, and 4 represent different donors. (**C**) Validation of the top miRNAs present in hMSCs. (**D**) Processes and pathways targeted by the most abundant miRNAs.

**Figure 3 biomedicines-10-00069-f003:**
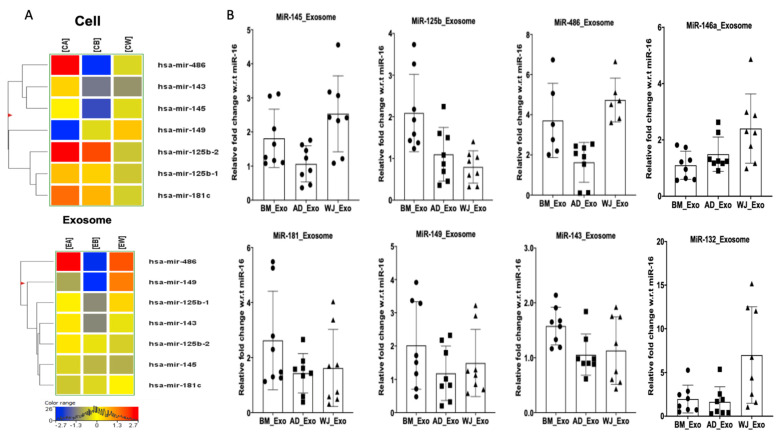
Expression patterns of miRNAs in hMSCs and exosomes. (**A**) Heatmap showing seven common miRNAs in hMSCs and exosomes. CB = Cells’ Bone marrow, CA = Cells’ Adipose tissue, CW = Cells’ Wharton Jelly; EA = Exosomes’ Adipose tissue, EB = Exosomes’ Bone marrow, EW = Exosomes’ Wharton Jelly. (**B**) Validations of the miRNAs in exosomes of three different sources. The miRNAs were isolated from exosomes and validated through qRT-PCR.

**Figure 4 biomedicines-10-00069-f004:**
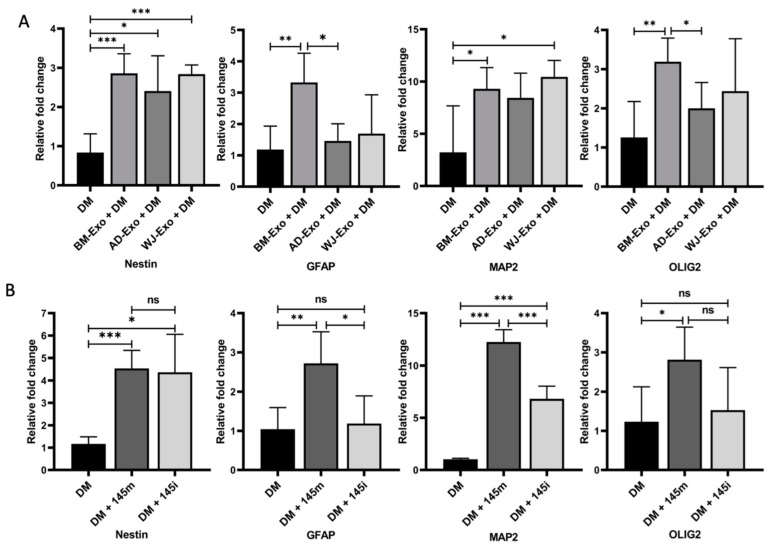
Effect of exosomes and miR-145 on neurogenesis. (**A**) C17.2 cells were incubated with differentiation media (DM) and 30 µg of exosomal protein for 7 days. Cells were lysed, and the expression of differentiation markers was measured using RT-PCR. (**B**) Cells were over-expressed and inhibited with mir-145-5p mimic or inhibitors. After 6 h of treatment, cells were washed and incubated with differentiation media for 7 days. Neurogenesis markers were checked by qRT-PCR. Data are represented as mean ± SD; * *p* < 0.033, ** *p* < 0.002, and *** *p* < 0.001; ns = non-significant; DM = Differential media; 145m = miR-145-5p Mimic; 145i = miR-145-5p inhibitor.

**Figure 5 biomedicines-10-00069-f005:**
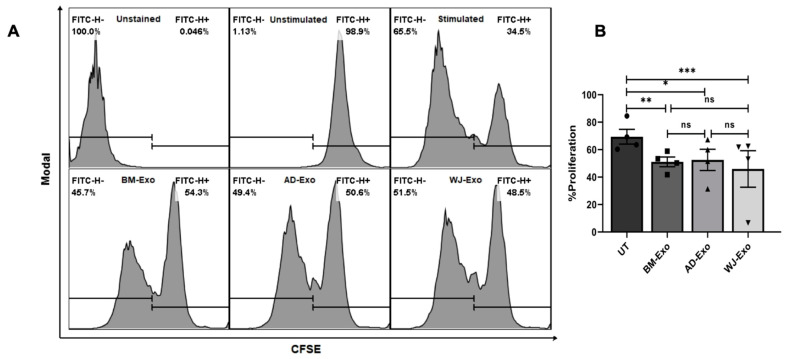
Effect of exosomes on PBMC proliferation. (**A**,**B**) PBMC isolated from the blood of healthy donors was used to study the proliferation assay. PBMC was stained with CFSE and incubated with 30 µg of exosomal protein for 6 days. The proliferation profile was checked through FACS analysis. (**B**) Percent proliferation is represented in the bar graph. Each dot represents one donor. Data are represented as mean ± SEM, * *p* < 0.033, ** *p* < 0.002, and *** *p* < 0.001; ns = non-significant.

**Figure 6 biomedicines-10-00069-f006:**
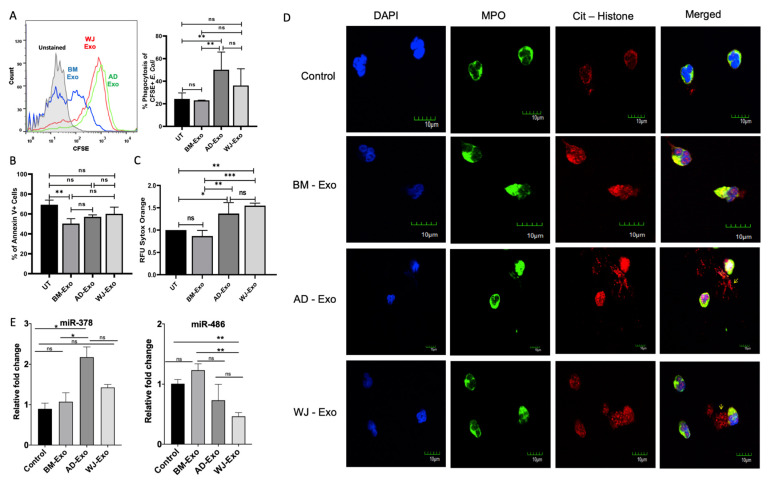
Effect of exosomes on neutrophil phagocytosis, apoptosis, and Neutrophil Extracellular trap formation. (**A**) A 30-µg equivalent exosomal protein for 8 h to study the effect on phagocytosis of *E. coli* cells. The percentage of CFSE-positive cells is represented in the bar graph. Data are represented as Mean ± SE. (**B**) The effect of exosomes on apoptosis of neutrophils was studied using Annexin V, and Propidium Iodide was used to differentiate early apoptotic cells and necrotic cells. The percent of the early apoptotic population from three experiments is shown in the bar graph. Data are represented as mean ± SD. (**C**) Sytox assay was performed to inspect the extracellular DNA release after 6 h of treatment with exosomes. Data are represented as mean ± SEM, * *p* < 0.033, ** *p* < 0.002, and *** *p* < 0.001. (**D**) Extracellular NET formation by neutrophils upon exosomes’ incubation for 3 h. Cells were stained with anti-MPO and anti-Cit-Histone antibodies and visualized under the confocal microscope. DAPI was used for nuclear staining. All cell images were captured with 40× objective. Scale bar = 10 µm. Arrow indicates release of NET in extracellular space. (**E**) Expression of miR-378a and miR-486 in neutrophils incubated with exosomes for 24 h. Bar plots show the mean ± SD; * *p* < 0.01 and ** *p* < 0.005.; ns = non-significant.

**Figure 7 biomedicines-10-00069-f007:**
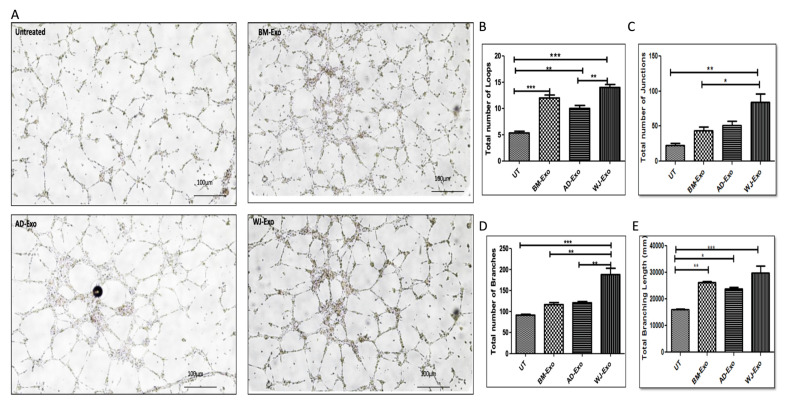
Human MSCs-derived exosomes enhance tube formation in HUVEC cells. (**A**) Formation of tube-like structure observed under a bright microscope: bar = 100 µm; Magnification = 4×. Bar graph representing (**B**) the total number of loops, (**C**) a total number of junctions, (**D**) a total number of branches, and (**E**) total branching length (in mm). WJ–Exo showed a significant increase in the tube formation by HUVEC cells in respect to untreated. Data are represented as mean ± SD, * *p* < 0.033, ** *p* < 0.002, and *** *p* < 0.001.

## Data Availability

All data generated or analyzed during this study are included in this published article. Data for small RNA sequencing are accessible under GEO series accession number GSE153752.

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
