# Peer review of "MicroRNA-Enriched Exosomes from Different Sources of Mesenchymal Stem Cells Can Differentially Modulate Functions of Immune Cells and Neurogenesis"

_biomedicines, 2021, doi:10.3390/biomedicines10010069_

Round 1
Reviewer 1 Report
In this manuscript titled “MicroRNA enriched exosomes from different sources of Mesenchymal Stem Cells can differentially modulate functions of immune cells and neurogenesis”, Soni et al characterized the miRNA pattern in exosomes of cultured hMSCs collected from three different tissue sources and tested the effect of exosomes on neurogenesis, in vitro T cell activation and expansion, neutrophil phagocytosis, NET formation, angiogenesis, etc. In this study, the authors clearly showed that the expression level of miRNA in exosomes had a different pattern from the expression level of miRNA in the parental hMSCs. And the miRNA level in exosomes also depends on the tissue source of the hMSCs, although the effect of exosomes from these different tissue sources in the functional assays tested had some similarities. The authors also showed that the hMSC derived exosomes might regulate neurogenesis through miR-145 by using miR-145 mimic and inhibitor. Overall, this is a thorough study on the exosomes and miRNA contained in exosomes, and this study showed a clear therapeutical potential of exosomes from different sources. However, there are several big logic gaps in the current manuscript which negatively affect the importance of this study. Especially, additional experiments are needed to establish that the miRNA, rather than other components such as proteins and lipids, in exosomes has a significant impact on the phenotypes the authors saw in these functional assays. If the authors can’t establish the link between miRNA in the exosomes and those exosome-mediated phenotypes, there is no point in analyzing the miRNA level in exosomes.
Major concerns:
- The authors did a thorough investigation of miRNA contained in the exosomes, and they showed that exosomes had effects on multiple physiological process such as neurogenesis, in vitro T cell activation and expansion, neutrophil phagocytosis, NET formation, angiogenesis through several functional assays. However, the authors have nearly zero evidence showing the miRNA of the exosomes has some impact in these assays, except for the single experiment performed on miR-145. Could the authors pick up several of the top miRNA expressed in these exosomes, and test their effect in those functional assays?
- Line 46 to 48, the authors demonstrated the safety concerns related with using hMSC for therapeutics, thus proposing the importance of using miRNA enriched exosomes. Instead of using miRNA enriched exosomes, why can’t we directly use miRNA for therapeutics? The authors probably want a better explanation and comparison of the advantages of using miRNA enriched exosomes versus using miRNA.
- In the figure 1C, the authors showed the expression pattern of several cell surface markers. However, these is no controls in these flow cytometry plots. The authors need at least some kind of control (non-staining, isotype control, expression on other cell population, etc.) to show the expression of these markers is real positive or real negative. Considering these flow cytometry plots may occupy a large space, the authors could consider moving the updated figure 1C with controls to supplemental figures.
- In figures 4 to 7, the authors only marked statistically significant difference among treatments, but didn’t show non-significant comparison, which will mislead readers. For example, in figure 5B, the graph showed there was statistically significant difference between UT and BM-Exo, as well as between UT and WJ-Exo, but readers don’t know whether there was statistically significant difference between UT and AD-Exo. When authors described the result of figure 5B, they said “BM and WJ-derived exosomes have a better suppressive capacity than AD-exosomes” (line 430-431). Does this mean the difference between UT and AD-Exo is not statistically significant? The authors need to clearly mark non-significant comparison as “ns” in the figure graph.
Minor concern:
The authors isolated hMSCs from different donors. In the current manuscript, it is unclear whether the authors separately cultured and tested these hMSCs from each individual donor or they pooled all the cells from donors together then split them for later experiment repeats. The authors need to clearly state it either in the method or in the figure legend, or have a separate supplement table to list the patient ID and/or sample ID used in each experiment
Reviewer 2 Report
Please find my comments and suggestions in the uploaded word file.

Round 2
Reviewer 1 Report
The statistic significance annotation in Figure 6E is still confusing, please annotate the statistic analysis result following the format of figure 6A-C.
All other concerns resolved.
Author Response
In the current version of the revised manuscript, we have redone statistical analysis as recommended by the reviewer and replaced Fig.6E with revised panel.